# Study on the Design, Preparation, and Performance Evaluation of Heat-Resistant Interlayer-Polyimide-Resin-Based Neutron-Shielding Materials

**DOI:** 10.3390/ma15092978

**Published:** 2022-04-19

**Authors:** Hu Xu, Dan Liu, Wei-Qiang Sun, Rong-Jun Wu, Wu Liao, Xiao-Ling Li, Guang Hu, Hua-Si Hu

**Affiliations:** 1College of Nuclear Science and Technology, Xi’an Jiaotong University, Xi’an 710049, China; xuhu_xu@126.com (H.X.); sunweiqiang@mail.xjtu.edu.cn (W.-Q.S.); 2Department of Radiation Protection and Detection, Sino Shaanxi Nuclear Industry Group, Xi’an 710054, China; 3The Sixth Research Laboratory, Wuhan Second Ship Design and Research Institute, Wuhan 430205, China; liudan@whhwtech.com (D.L.); wurongjun@whhwtech.com (R.-J.W.); liaowu@whhwtech.com (W.L.); lixl005@163.com (X.-L.L.)

**Keywords:** neutron shielding, heat resistance performance, optimal design, lightweight interlayer

## Abstract

Polymers have an excellent effect in terms of moderating fast neutrons with rich hydrogen and carbon, which plays an indispensable role in shielding devices. As the shielding of neutrons is typically accompanied by the generation of γ-rays, shielding materials are developed from monomers to multi-component composites, multi-layer structures, and even complex structures. In this paper, based on the typical multilayer structure, the integrated design of the shield component structure and the preparation and performance evaluation of the materials is carried out based on the design sample of the heat-resistant lightweight polymer-based interlayer. Through calculation, the component structure of the polymer-based materials and the three-layer thickness of the shield are obtained. The mass fraction of boron carbide accounts for 11% of the polymer-based material. Since the polymer-based material is the weak link of heat resistance of the multilayer shield, in terms of material selection and modification, the B_4_C/TiO_2_/polyimide molded plate was prepared by the hot-pressing method, and characterization analysis was conducted for its structure and properties. The results show that the ball milling method can mix the materials well and realize the uniform dispersion of B_4_C and TiO_2_ in the polyimide matrices. Boron carbide particles are evenly distributed in the material. Except for Ti, the other elemental content of the selected areas for mapping is in good agreement with the theoretical values of the elemental content of the system. The prepared B_4_C/TiO_2_/polyimide molded plate presents excellent thermal properties, and its glass transition temperature and initial thermal decomposition temperature are as high as 363.6 °C and 572.8 °C, respectively. In addition, the molded plate has good toughness performs well in compression resistance, shock resistance, and thermal aging resistance, which allows it to be used for a long time under 300 °C. Finally, the prepared materials are tested experimentally on an americium beryllium neutron source. The experimental results match the simulation results well.

## 1. Introduction

The material exerts its shielding effect through an interaction with neutrons and γ-rays by its internal nuclei and extra-nuclear electrons, respectively. For neutrons, the smaller atomic number the nuclide, the better the moderating effect, which makes polymers indispensable in shielding devices. In the early days, shielding materials were often monomer materials such as water, polyethylene, paraffin, and lead. As with the development of nuclear technology application devices, the diversified functions, compactness, light weight, and high-power results constitute increasingly strict requirements for shielding materials. Shielding materials were gradually developed into multi-component composites or multi-layer structures, or even more complex coating structures [1,2].

The requirement for shielding materials varies based on different types of nuclear radiation. Low-density materials such as PMMA, leaded glass, rubber products, fiber fabrics, and concrete are more often used to shield low-energy X-rays [3]; heavy metals such as iron, lead, and tungsten, as the reinforcing phase particles, are effective choices for γ-rays [4,5], and lead may also be replaced with alternative materials due to its chemical toxicity [6]; materials with high hydrogen content such as paraffin, polyethylene, and polypropylene are more effective as fast neutron moderator materials to shield neutrons. Materials containing boron or lithium elements such as elemental boron, boric acid, boron carbide, lithium bromide, and lithium fluoride can be used as excellent slow neutron absorbers. When the neutron energy is very high, a layer of heavy metal is often placed at the very front to reduce the neutrons’ energy through inelastic scattering. A single material cannot protect against the coupling of multiple kinds of nuclear radiation in reality. Therefore, it is essential to develop a composite material that can shield multiple radiations. In such a material, heavy substances would shield X-rays, γ-rays, etc., and have the effect of moderating high-energy neutrons through inelastic scattering, while light elements would be mainly used to further slowdown the neutrons.

Due to their unique molecular structure and high content of hydrogen for neutron moderation, polymer materials are indispensable in shields. However, compared with metal and ceramic materials, polymer-based materials are often the bottleneck to heat the resistance performance of the entire shield. Therefore, the development of high-temperature-resistant polymer-based nuclear radiation shielding materials has become the priority. In early days, polyethylene was widely used since would not be activated due to its high hydrogen content. However, it softens at 110 °C with poor radiation resistance performance, which means it must be replaced before becoming invalid because of decomposition by radiation-mixed boron carbide and ultra-high-molecular-weight polyethylene if one seeks to develop a new neutron composite shielding material and an added KH570 silane coupling agent to improve the interface effect of the material. The neutron attenuation coefficient of the material improves with the increase of boron carbide content, but excessive particles will cause increased agglomeration and reduce the shock resistance of the material [7]. In addition, NASA studied boron-containing multifunctional composites, and they chose high-density polyethylene as the continuous phase and boron carbide and boron nitride as reinforcing fibers. High-density polyethylene outperforms pure polyethylene in tensile modulus, tensile strength, and shielding effect [8].

However, the weakness in heat resistance and radiation resistance of polyethylene-based material have limited its use in many environments. Therefore, scholars have carried out research on other composite materials with other bases, such as epoxy resin and hydrogel, and yielded a great deal of scientific research achievements. The Japan Institute of Technology has designed a new type of neutron-shielding material that uses epoxy resin as the base and colemanite as the reinforcing phase. Compared with concrete and polyethylene boron carbide composite materials, it enhances mechanical properties, achieves heat resistance temperatures as high as 133 °C, and presents a better shielding effect [9]. JAEA developed a series of epoxy resin and heat-resistant neutron composite materials with different contents of boron. The developed material performs well in both shielding and heat resistance and can be used in the external shield of vacuum containers and superconducting Tokomak device [10]. Seyhun Kipcak has studied epoxy-based composites with different contents of boron fiber. Those materials are based on epoxy resin and are configured according to different combinations of boron fiber and boron nano powder B_4_C, which perform excellently in terms of both their shielding effect and mechanical properties [11]. Kyoto University developed a series of samples with thermosetting resin as the base. It has been verified that among all the samples, the resin + 35% ^6^LiF composite material is the most effective and practical neutron-shielding material, and it can be applied in medical and biological facilities such as boron neutron capture therapy [12]. Masoud Sabzi and Morteza Shamanian developed a series of new metal matrix composite shielding materials and achieved good results [13,14,15]. Guang Hu developed a high-temperature-resistant composite by adding boron carbide to epoxy resin [16]. PVA/PEO hydrogel materials were successfully developed by physical cross-linking method and based on which new metal ion-containing hydrogel materials were developed by adding compounds containing heavy metal ions and rare earth elements [17,18]. Additionally, the micro and nanostructured composite materials for neutron-shielding applications was reviewed by Sajith [19]. Polymer-based materials have great potential in high temperature resistance, and the improvement of their heat resistance is the key to improving the service performance of multilayer shields. This paper studies the design, preparation, and performance evaluation of polymer-based interlayer materials based on the structural optimization of typical multilayer structures.

## 2. Material and Method

### 2.1. Optimal Design of the Materials

The composition of the materials is designed as Figure 1 prior to material preparation. The source adopts the fission neutron spectrum and fission gamma ray spectrum, which are vertically incident to the shield from the left side. The shield has a total of three layers, which are 304 stainless steel (SS), a polymer-based composite material, and 304 stainless steel (SS), with the polymer-based material as the middle interlayer. The polymer-based material is composed of polyimide and boron carbide. In this way, the middle interlayer is lightweight and easy to prepare. The iron on both sides is a traditional material. The block shied, with a total thickness, height, and width of 10 cm, can provide better support. At far right is a detector with a height and width both of 10 cm. The multi-layer shield is modeled and calculated by MCNP software, and the shield is optimized by combining with AI algorithms to obtain the composition of the polymer-based material and the thickness of each layer of the three-layer shield.

The designed method is established by genetic algorithm (GA) combing with MCNP [20]. The parameters, such as the thickness, density, and components of the material, are set in cell cards and material cards in the MCNP code. The data representative of the dose equivalent of neutrons and γ rays in the output file are extracted. The extracted data are set as the objective function of GA. In the GA program, the optimal combination of structure, components, and density is sought out for satisfying a preset objective. In the shielding design, the objective commonly is the lowest dose equivalent of neutrons and γ-rays after their penetration through the shield. Equation (1) is the objective function
Min*H* (*L*, *A*) = min [*αH_n_*(*L*, *A*) + *βH_γ_*(*L*, *A*)](1)
where $H_{n}(L,A)$ is the dose equivalent of the neutron and $H_{\gamma }(L,A)$ is the dose equivalent of γ ray.α represents the counting of neutrons, and β represents counting of γ-rays in the initial source. Equations (2)–(4) are the constraint conditions.
(2)$$\sum_{i=1}^{p}\frac{L_{i}}{L_{all}}=1$$
(3)$$\sum_{i=1}^{p}A_{1}+A_{2}+\cdot \cdot \cdot A_{i}=1$$
(4)$$\rho_{x}\leq \rho_{eff}\leq \rho_{y}$$
where $L_{i}$ is the thickness of each layer, $L_{all}$ is the total thickness of the shield, $A_{i}$ is the mass ratio of each component, and $\rho_{eff}$ is the equivalent density of the shield.

Figure 2 is the flow chart of GA combing with MCNP, which is used to design the structure and component of the shield. It contains five steps.

(1)Input the parameters expressing the thickness of the shield and the components of the materials.(2)Produce the “inpn” file and “inpp” file for simulating the neutron and γ ray transmission in the material.(3)The “inpn” file and “inpp” file are calculated by MCNP, and the “outpn” file and “outpp” are produced.(4)Extract the data expressing the dose equivalent of neutron in the “outpn” file and the γ-rays in the “outpp” file.(5)The program stops when the fitness value is not changed or the iteration times reach to N_0_ (the generation number). If not, a new thickness and component will be produced and then the next calculation will start.

GA can seek out the optimal combination solutions of a question quickly from all combinations. In our established method, the function of GA is seeking out the optimal combination of thickness and the components from all the combinations. With generation after generation’s calculation of the GA method combing with MCNP, the total dose equivalent of neutrons and γ rays tends to be small on the whole. Figure 2 shows the relationship between them.

Through calculation, the component structure of the polymer-based material and the three-layer thickness of the shield are obtained. The mass fraction of boron carbide accounts for 11% of the polymer-based material. The shielding effect comparison between the designed material and several typical combined shields are as shown in Figure 3, in which the group (1) SS + 43.5 vol% composite + SS is the designed sample, the thickness of the first layer is SS, the second layer of the polymer-based material is 4.35 cm thick, and the third layer material is 0.42 cm thick; the group (2) SS + 43.5% sphere composite seeks to make the polymer-based material into a ball, the radius of which is 0.153 cm, and the total thickness of the material is 10 cm; the group (3) SS + 50.6% sphere B_4_C seeks to make B_4_C into a ball, the radius of which is 0.153 cm, the total material thickness is 0.153 cm, and the thickness is 10 cm; group (4) BSS is boron-containing stainless steel with a mass fraction of B_4_C of 1.7%; group (5) is a boron carbide material; and the group (6) composite is a designed polymer-based interlayer material. It can be seen from the results that the designed samples are comparable to groups (5) and (3), which are better than the other groups. Additionally, the designed sample has the advantages of simple disassembly and installation, and good support structure. Moreover, it just needs to prepare the middle interlayer.

### 2.2. Preparation of Materials

The main raw materials used in this experiment are as follows: (1) the thermosetting polyimide molding powder; (2) the boron carbide (B_4_C) powder, 99.9%; (3) the nano titanium dioxide (TiO_2_) powder, which is analytically pure. Here, thermosetting polyimide molding powder is mainly used for fast neutron moderation; boron carbide (B_4_C) powder is mainly used for thermal neutron absorption, and for fast neutron moderation, but its moderation ability is slightly inferior to that of thermosetting polyimide molding powder; and nano titanium dioxide (TiO_2_) is mainly employed for material modification. All raw materials are dried at 200 °C for 2 h before use.

The key factors influencing the performance of the B_4_C/TiO_2_/polyimide molded plate mainly include the raw material properties, the uniformity of component distribution, and the molding process. Therefore, two aspects are focused on during the preparation of the molded plate: (a) even mixing of the molding powder; and (b) the hot pressing process.

(a) Blending of raw materials

In order to ensure the even dispersion of B_4_C and TiO_2_ in the polyimide matrix, B_4_C, TiO_2,_ and the polyimide molding powder need to be thoroughly compounded before hot pressing. Here, the MSK-SFM-1-1L planetary ball mill is selected to blend B_4_C, TiO_2,_ and the polyimide molding powder. The details are as follows: firstly, pretreat polyimide molding powder and B_4_C and TiO_2_ powders at 200 °C for 2 h; secondly, put polyimide molding powder, B_4_C powder, TiO_2_ powder, and ceramic balls into the ceramic ball mill jar, according to 87:10:3:50 (mass ratio), and then place the jar into the ball mill; thirdly, turn on the ball mill and set the rotational speed to 380~500 r/min, then turn off the ball mill after 30~60 min of ball milling; and finally, use a 100-mesh sieve to separate the ceramic balls and obtain the well-mixed B_4_C/TiO_2_/polyimide powder.

In this process, not only can B_4_C, TiO_2,_ and the polyimide be evenly mixed by ball milling but the particle size and distribution of the mixed powder can be controlled by adjusting the diameter and proportion of the ceramic balls, which lays a good foundation for the preparation of the B_4_C/TiO_2_/polyimide molded plate with excellent performance.

(b) Hot-pressing process

The hot-pressing process of the polyimide profile is the key factor that affects the performance of its products. During the experiment, after preliminary process exploration and optimization, the hot-pressing process of the B_4_C/TiO_2_/polyimide molded plate is determined as follows.

Weigh 140 g of the mixed molding powder and put it into the customized hot-pressing mold, and then place it between the upper and lower platens of a hot press machine; after that, perform cold pressing on the mold at 30 MPa and then release the pressure, and set the temperature of the upper and lower platens at the same time to 300 °C. Preserve the heat for 30 min when the temperature reaches 300 °C, and then pressurize the mold to more than 20 MPa; afterwards, depressurize and repeatedly exhaust the air; after exhausting, pressurize the upper and lower platens to 20 MPa, and set the temperature of them at the same time to 390 °C. When the temperature reaches 390 °C, preserve the heat for 20 min, and then stop heating; when the temperature of the platens drops to 200 °C, take out the mold and demold the sample, and a B_4_C/TiO_2_/polyimide molded plate with good toughness can be obtained. Figure 4 shows the pressed B_4_C/TiO_2_/polyimide molded plate, the size of which is 100 mm × 100 mm × 10 mm.

## 3. Results and Discussion

### 3.1. Analysis of Chemical Structure and Composition

FT-IR and XPS are used to characterize and analyze the chemical composition, structure, and element valence of the B_4_C/TiO_2_/polyimide molded plate. Figure 5 shows the FT-IR spectra of the molding powder before hot pressing and the molded plate after hot-pressing. As can be seen, the spectra show obvious characteristic absorption peaks of the imine ring, before and after the sample is hot pressed, at 1780 cm^−1^ (*γ*_C = O_, asymmetric stretching vibration), 1725 cm^−1^ (*γ*_C = O_, symmetric stretching vibration), 1375 cm^−1^ (*γ*_C-N_, stretching vibration), and 720 cm^−1^ (*γ*_C = O_, bending vibration), indicating that the imine ring structure of the polyimide will not be destroyed after hot pressing. After hot pressing, the C-N bond stretching vibration characteristic peak of polyimide at 1375 cm^−1^ red-shifts (i.e., moves to the low wavenumber direction), with an offset of about 9 cm^−1^. The reason might be, after hot pressing, the polyimide molecular chains are crosslinked, so the rigidity of the molecular chains is further increased; additionally, the imine ring is twisted to a certain extent; the C-N bond energy in the ring is slightly reduced; and the polyimide molecular stability is decreased. These lead to the red-shift of the corresponding stretching vibration peak.

### 3.2. Microscopic Morphology Test and Analysis

SEM was used to analyze the microstructure in the B_4_C/TiO_2_/polyimide molded plate and the distribution of B_4_C and TiO_2_. Before the test, the molded plate was ruptured by applying an external force to expose its internal microstructure. In order to show the distribution of B_4_C and TiO_2_ more clearly, we selected two different areas for analysis. Figure 6 and Figure 7 show the SEM photos of the cross-section of the molded plate.

In Figure 6 and Figure 7, the B_4_C particles are in red circles and the TiO_2_ particles are in green circles. It can be seen from the figures that B_4_C and TiO_2_ are evenly dispersed in the polyimide matrix. B_4_C is irregularly shaped, with poor uniformity of particle size.

The large particle size is around 30–50 μm, and the small particle size is around 5 μm. The TiO_2_ particles are small and adhere to the polyimide matrix, which shows good compatibility. The polyimide matrix in the cross-section has a sheet-shaped porous structure, which may be due to the ductile fracture of the polyimide resin and the “wire pulling”-like phenomenon when it is damaged by an external force.

The distribution of B_4_C and TiO_2_ in the B_4_C/TiO_2_/polyimide molded plate was further confirmed by an energy spectrometer, and the test results are shown in Figure 8. According to the components and the elemental composition of the molded plate, the distribution of the B element is the key to confirming B_4_C, and the Ti element is the key to confirming TiO_2_ distribution. The distribution of B element matches well with the black masses in Figure 3c (the first figure), further proving that the black masses are the added B_4_C particles. Figure 8c (the fifth figure) shows the distribution of the Ti element, and the Ti elements are more evenly distributed but with a low content, which is mainly due to a low TiO_2_ addition, smaller particles, better compatibility with the polyimide matrix, and heavy embedding. Combined with the analysis results in Figure 6, Figure 7 and Figure 8, it is strongly proved that B_4_C and TiO_2_ have been evenly dispersed in the polyimide matrix. Table 1 shows the content of the elements in the system and their theoretical values obtained from the cross-sectional mapping test. It is clear that the elemental content of the selected areas for mapping is in good agreement with the theoretical values of the elemental content of the system, except for Ti. There are two main reasons for the large error of the Ti element: the content of Ti element is low; therefore, a small measurement error will lead to a large error in the Ti element content; and TiO_2_ is well embedded by the polyimide resin, which makes some Ti elements undetectable.

### 3.3. Thermal Properties Analysis

The thermal properties of the B_4_C/TiO_2_/polyimide molded plate are characterized and analyzed by adopting DMA and TGA. Figure 9 shows the process of the DMA test on the molded plate, and its results. According to the Figure, as the temperature rises, the storage modulus of the molded plate presents certain fluctuations under a slow rising trend, while the degree of change is not significant. When the temperature reaches about 240 °C, the storage modulus decreases slowly, which might be caused by the secondary relaxation phenomenon due to the “thawing” of small motion units of the polyimide molecular chains. When the temperature reaches about 300 °C, an obvious turning point of the storage modulus-temperature curve appears with an accelerated decrease, illustrating that the polyimide molecular chains have gradually thawed and have become “free”. When the temperature reaches about 400 °C, the decrease of the storage modulus becomes slow again, indicating that the polyimide molecular chains have completed “thawing” and become fully “free”. In consideration of the wide thawing temperature range of the polyimide molecular chains, the peak of the Tanδ- temperature curve within this range is selected as the glass transition temperature, which means the glass transition temperature of the B_4_C/TiO_2_/polyimide molded plate is 363.6 °C.

Figure 10 shows the TGA-DTG curve. To effectively reduce the influence of the thermal hysteresis on the TAG test, we grind the plate into powder for the test. The TAG curve proves the outstanding thermal stability of the prepared molded plate. When the temperature is below 550 °C, no significant weight loss occurs. With the initial pyrolysis temperature as high as 572.8 °C, the molded plate’s *T*_d_, 5% (temperature corresponding to 5% weight loss) and *T*_d_, 10% (temperature corresponding to 10% weight loss) are 582.7 °C and 607.8 °C, respectively. At 800 °C, the char formation rate is 23.24 wt%. It can be seen from the DTG curve that the temperature for the maximum pyrolysis rate is 617.2 °C, meaning the plate reaches the maximum pyrolysis rate at 617.2 °C.

### 3.4. Mechanical Properties Analysis

Adding additives to a polymer is an important method to improve a polymer’s properties and endow it with special characteristics. In this process, the addictive type, volume of addition, and scattering status influence the mechanical properties of the compound materials in a significant way. Considering the B_4_C/TiO_2_/polyimide molded plate will be used for a long time under a high temperature of 300 °C, the molded plate is aged for 100 h at 300 °C, and its compression and impact resistance before and after aging are tested. To ensure the accuracy of test results, the compression and impact resistance are tested three times to assess the mechanical properties and anti-aging properties of the molded plate accordingly.

Figure 11 shows the compression test process and stress-strain curve of the B_4_C/TiO_2_/polyimide molded plate. The compression rate of the experiment is 1 mm/min., and the environment temperature is 10 °C. According to the stress-strain curve in the Figure 11a), at the initial compression of three tests (deformation lower than 4%), the linear change presented by the stress-strain curve indicates the plate is in the elastic deformation process of the compression. The calculated Young’s modulus in three tests is 2.65 GPa, 2.61 GPa, and 2.38 GPa, respectively, with an average value of 2.55 ± 0.12 GPa. Later, the increase of stress gradually slows and basically remains unchanged after the deformation exceeds 10%. The maximum compression strength for the three tests is 125.39 MPa, 121.95 MPa, and 128.68 MPa, with an average value of 125.34 ± 2.75 MPa. When the deformation is less than 8%, no obvious change occurs on the surface of the sample. However, when the deformation exceeds 8%, cracks gradually appear on the molded plate’s surface. There is no significant brittle fragmentation phenomenon in the compression process, which is further proof of the soundness and toughness of the B_4_C/TiO_2_/polyimide molded plate. Therefore, it follows that the B_4_C/TiO_2_/polyimide molded plate presents outstanding compression resistance properties.

For the stress-strain curve after aging, there is no significant trend of weakening in terms of change and peak. In initial compression, the curve also presents linear change, and the Young’s modulus is 2.32 GPa, 2.36 GPa, and 2.30 GPa, respectively, making for an average value of 2.33 ± 0.03 GPa. It is revealed that after the thermal aging, the Young’s modulus of the molded plate decreases slightly, meaning the aging treatment reduces the rigidity of the molded plate. The maximum compression strength for three compressions after aging is 124.68 MPa, 127.19 MPa, and 132.30 MPa, respectively, with an average value of 128.06 ± 3.17 MPa. The results, which demonstrate that the maximum compression strength after aging is even slightly higher than that before the treatment, indicate that the thermal aging treatment will not produce adverse influences on the compression resistance properties of the molded plate. Table 2 shows the test results of the impact resistance properties of the molded plate before and after aging. According to the results, a good impact resistance properties is presented by the average impact resistance strength of 11.77 ± 0.59 kJ/m^2^ before the aging; after the test, the average impact resistance strength is 11.57 ± 0.58 kJ/m^2^, a slight decrease of 1.3% compared with that before the aging. The raw material polyimide is thermosetting. The molecular bond breaks at high temperature, which gradually reduces its mechanical properties. The results show that the material can be used long-term under 300 °C.

### 3.5. Shielding Performance Test

The shielding experiment of the polymer-based interlayer material was carried out on the americium beryllium neutron source. The experimental layout is shown in Figure 12. During the actual measurement, a neutron dose equivalent meter was used to measure the dose equivalent behind the shielding material.

The shadow cone can completely block normal-incident neutrons and is mainly used to measure the dose of scattered neutrons. The polyethylene moderator layer mainly moderates the americium beryllium neutron source to measure the shielding effect under the two neutron energy spectra. Figure 13 shows the experimental and simulation results of the polymer-based materials, and experiments and simulations are carried out on boron-containing steel with a boron content of 1.7%. It can be seen from the results in Table 3 that the experimental results are in good agreement with the simulation results, which verifies the accuracy of the design and preparation. The neutron-shielding effect of the polymer-based interlayer material is inferior to that of the boron-containing steel because the average energy of the americium-beryllium neutron source is 4.5 MeV, and the moderating effect of the iron is dominant in this energy range. The polymer-based interlayer material is more effective with the help of the outer layer steel.

## 4. Conclusions

The component structure of the polymer-based materials and the three-layer thickness of the shield are obtained by the design of GA combined with MCNP. The mass fraction of boron carbide accounts for 11% of the polymer-based materials. The B_4_C, TiO_2,_ and polyimide molding powder are blended via ball milling and made into a B_4_C/TiO_2_/polyimide molded plate through hot pressing. The density and composition of the materials were tested. The elemental content of the selected areas for mapping is in good agreement with the theoretical content. Then, the characterizations and analysis of its structure and properties are performed. The results show that B_4_C and TiO_2_ are evenly dispersed in the polyimide matrix, proving that the materials are well-blended by ball milling. The prepared B_4_C/TiO_2_/polyimide molded plate presents outstanding thermal properties, with its glass transition temperature and initial pyrolysis temperature up to 363.6 °C and 572.8 °C, respectively. The test results of the impact resistance properties of the molded plate before and after aging. According to the results, a good impact resistance property is presented by the average impact resistance strength of 11.77 ± 0.59 kJ/m^2^ before the aging; after the test, the average impact resistance strength is 11.57 ± 0.58 kJ/m^2^, a slight decrease of 1.3% compared with that before the aging. In addition, the plate demonstrates sound toughness, and excellent performance on compression resistance, impact resistance, and thermal aging resistance, which allow for its long-time use under 300 °C. Finally, the prepared materials are tested experimentally on an americium beryllium neutron source. The experimental results match the simulation results well.

## Figures and Tables

**Figure 1 materials-15-02978-f001:**
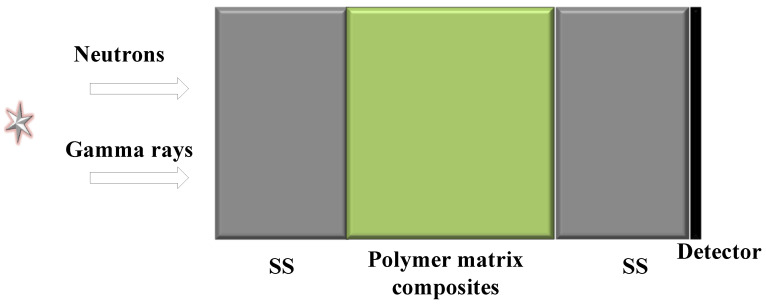
Multilayer structural material containing a polymer-based interlayer.

**Figure 2 materials-15-02978-f002:**
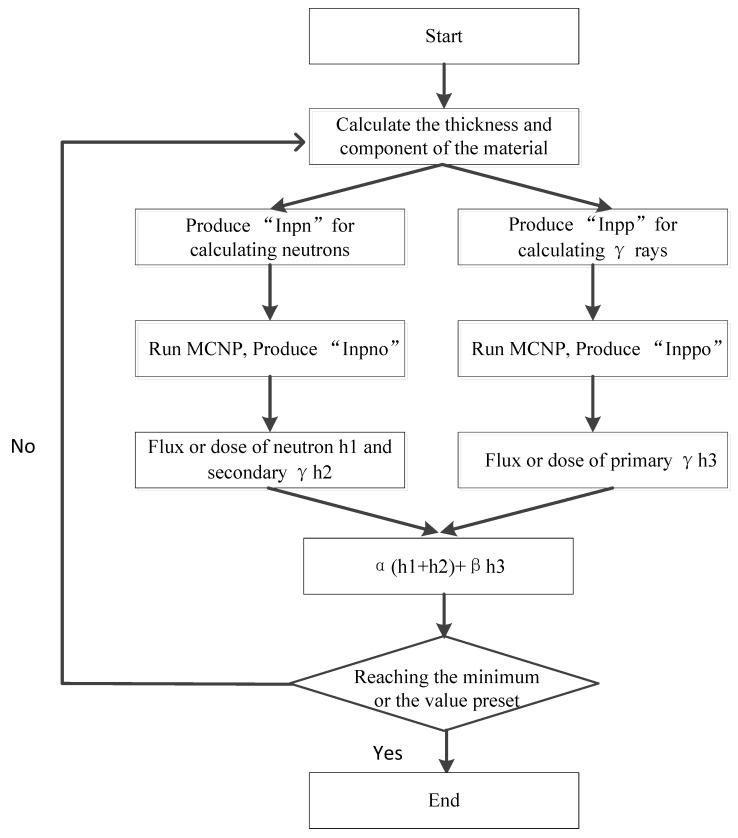
Flow chart of representing the structure and components of the shielding.

**Figure 3 materials-15-02978-f003:**
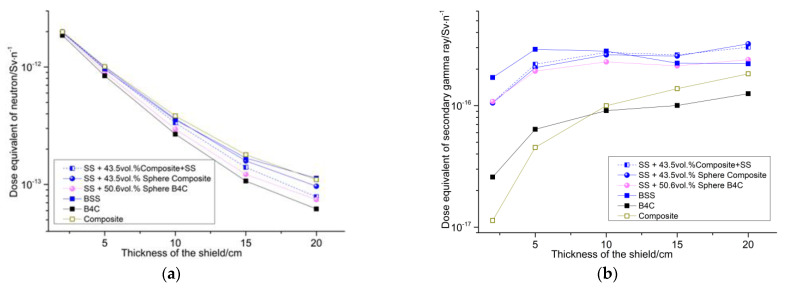
The shielding performance of six groups of materials. (**a**) The dose equivalent of the neutron. (**b**) The dose equivalent of the second gamma rays. (**c**) The dose equivalent of the primary gamma rays. (**d**) The dose equivalent of the total.

**Figure 4 materials-15-02978-f004:**
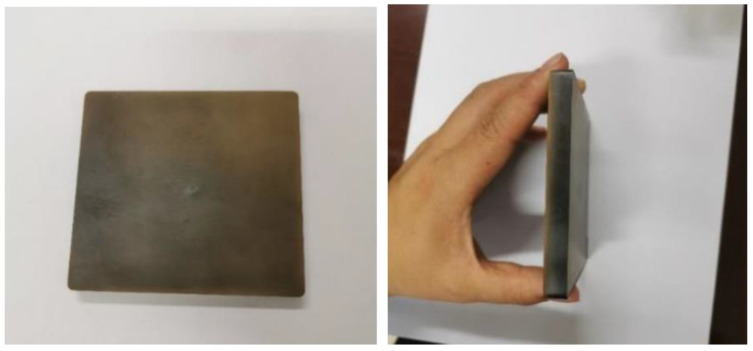
The B_4_C/TiO_2_/polyimide molded plate.

**Figure 5 materials-15-02978-f005:**
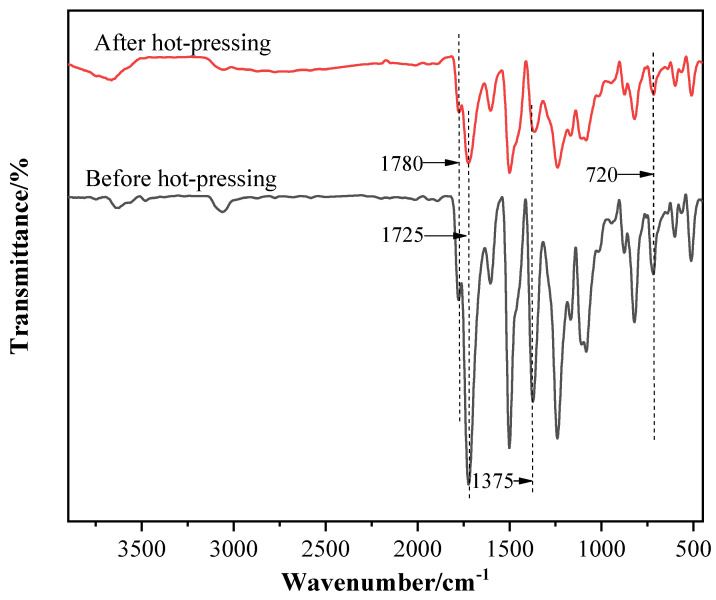
The FT-IR spectra of the molding powder before hot pressing and the molded plate after hot pressing.

**Figure 6 materials-15-02978-f006:**
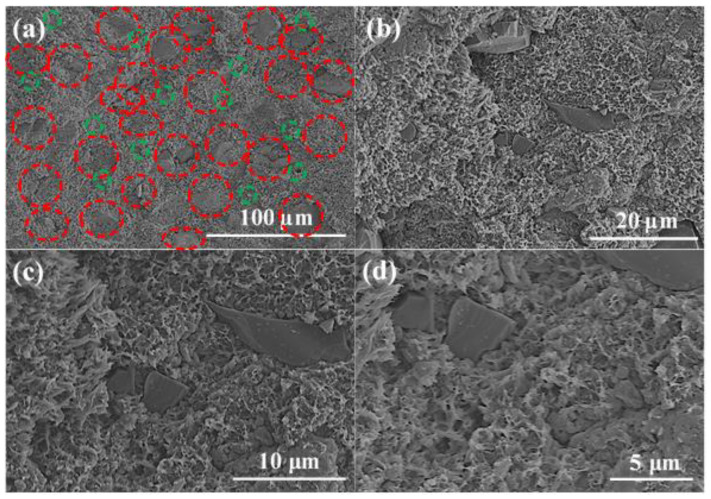
SEM photo of the cross-section of the B_4_C/TiO_2_/polyimide molded plate (Area 1). (**a**) B_4_C, TiO_2_ particle distribution (100 μm), (**b**) B_4_C, TiO_2_ particle distribution (20 μm), (**c**) B_4_C, TiO_2_ particle distribution (10 μm), (**d**) B_4_C, TiO_2_ particle distribution (5 μm).

**Figure 7 materials-15-02978-f007:**
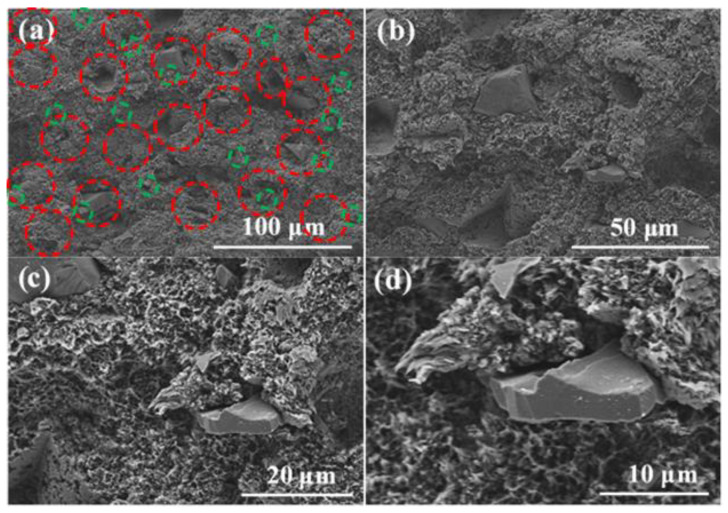
SEM photo of the cross-section of the B_4_C/TiO_2_/polyimide molded plate (Area 2). (**a**) B_4_C, TiO_2_ particle distribution (100 μm), (**b**) B_4_C, TiO_2_ particle distribution (50 μm), (**c**) B_4_C, TiO_2_ particle distribution (20 μm), (**d**) B_4_C, TiO_2_ particle distribution (10 μm).

**Figure 8 materials-15-02978-f008:**
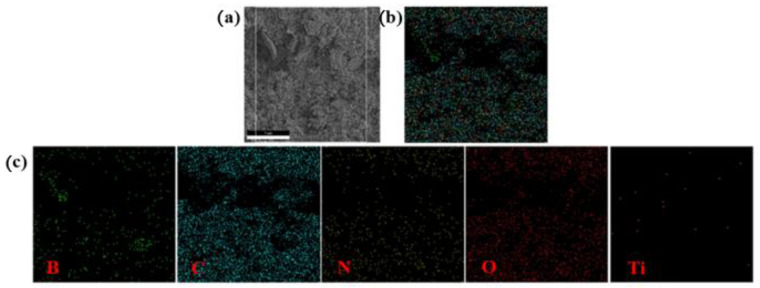
(**a**) An SEM photo of the material; (**b**) the element distribution map; and (**c**) EDS elemental mapping consisting of B, C, N, O, and Ti.

**Figure 9 materials-15-02978-f009:**
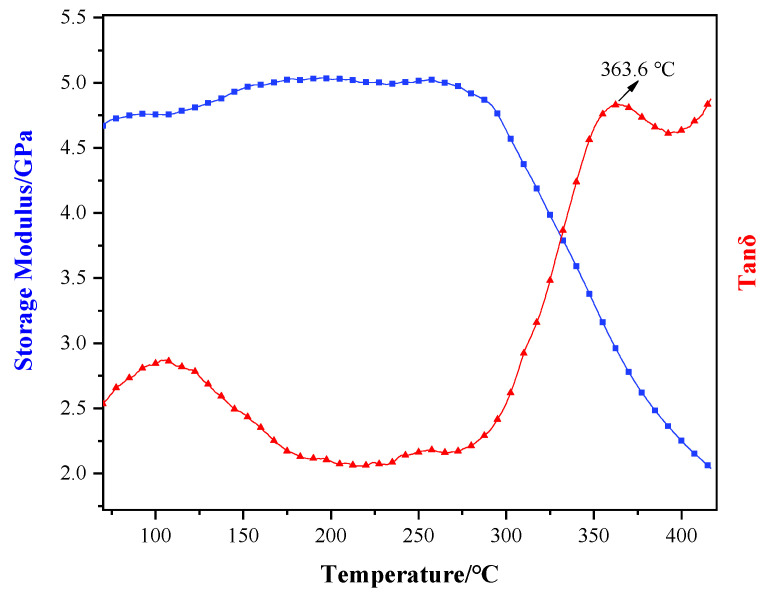
The storage modulus-temperature curve (blue) and the Tanδ-temperature curve (red) of the B_4_C/TiO_2_/polyimide molded plate.

**Figure 10 materials-15-02978-f010:**
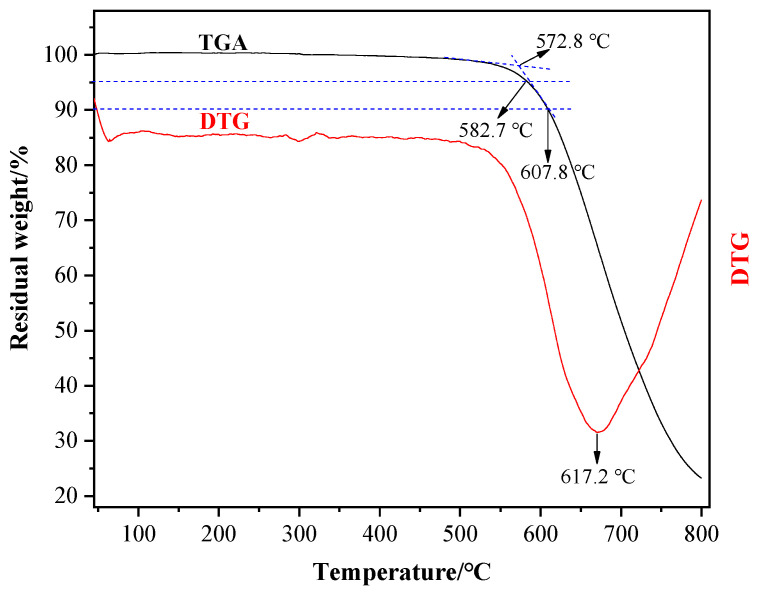
The TGA-DTG curve of the molded plate.

**Figure 11 materials-15-02978-f011:**
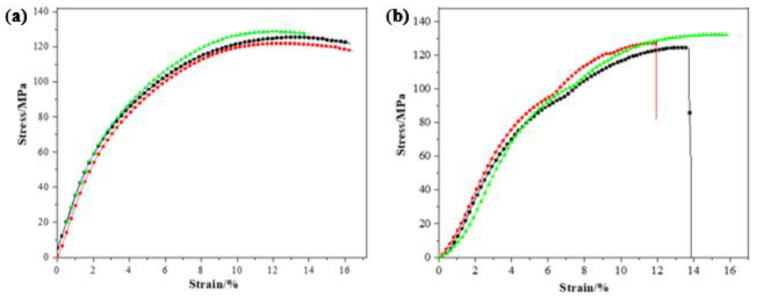
The strain-stress curve of the molded plate: (**a**) before aging and (**b**) after aging.

**Figure 12 materials-15-02978-f012:**
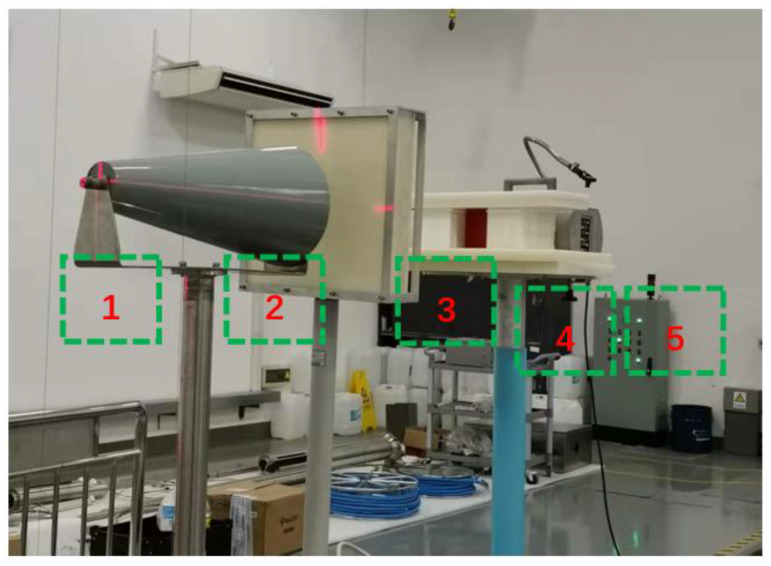
The neutron-shielding experimental layout (1: the neutron source; 2: the shadow cone; 3: the polyethylene moderator layer; 4: the polyethylene shield layer (to shield incident neutron); and 5: the neutron dose equivalent meter).

**Figure 13 materials-15-02978-f013:**
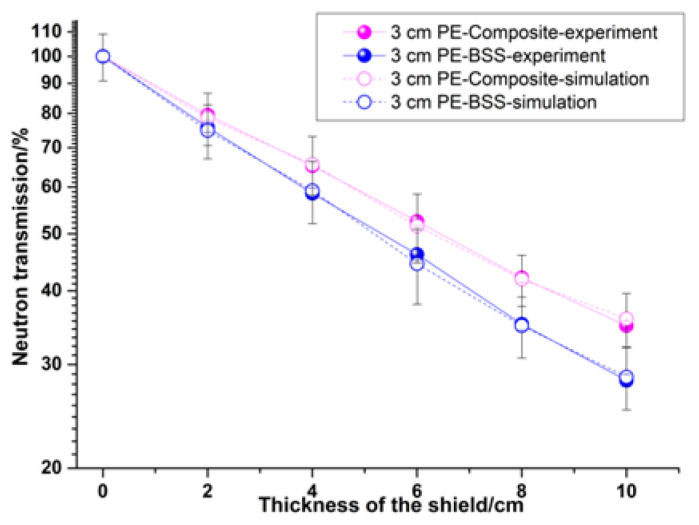
A comparison of the experimental and simulation results of two materials.

**Table 1 materials-15-02978-t001:** The elemental content of the B_4_C/TiO_2_/polyimide molded plate cross-section.

Element	B	C	N	O	Ti
Measured value (wt%)	7.38	62.21	10.90	18.48	1.03
Theoretical value (wt%)	7.82	62.30	9.54	16.84	1.80

**Table 2 materials-15-02978-t002:** The impact resistance properties of the B_4_C/TiO_2_/polyimide molded plate.

Test		Section Width (mm)	Section Thickness (mm)	Absorption Work (J)	Impact Resistance Strength (kJ/m^2)^
Before Aging	1	10.25	3.57	0.42	11.48 ± 0.57
	2	10.24	3.34	0.41	11.99 ± 0.60
	3	10.22	3.64	0.44	11.83 ± 0.59
	Average	-	-	-	11.77 ± 0.59
After Aging	1	10.16	3.37	0.40	11.68 ± 0.58
	2	10.22	3.59	0.42	11.45 ± 0.57
	3	10.15	3.49	0.41	11.57 ± 0.58
	Average	-	-	-	11.57 ± 0.58

**Table 3 materials-15-02978-t003:** The simulation and experimental results of two materials at 10 cm.

Material	Composite (10 cm)		BSS (10 cm)	
-	Experiment	Simulation	Experiment	Simulation
Transmission (%)	34.88	35.87	28.18	28.56

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
