# Peer review of "Study on the Design, Preparation, and Performance Evaluation of Heat-Resistant Interlayer-Polyimide-Resin-Based Neutron-Shielding Materials"

_materials, 2022, doi:10.3390/ma15092978_

Round 1

Reviewer 1 Report

The paper contains some interesting results that make it publishable after the following mandatory revisions:

1-The title of the paper is incomplete. Details such as type of the interlayer should be mentioned in the title of the paper.

2-Abstact is incomplete. Extra sentences should be deleted. The parameters investigated and the tests used and the general results should be motioned in this section.

3-In the introduction of the paper, layer deposition processes and their effect on the properties of the layer should be discussed. The current introduction is incomplete. To modify this section, the following references can be used:

- Deposition of ceramic nanocomposite coatings by electroplating process: a review of layer-deposition mechanisms and effective parameters on the formation of the coating, Ceramics International, Vol. 45 (17), 2019, 21835-21842.

- CLAD ALUMINUM ALLOY PRODUCTS AND METHODS OF MAKING THE SAME, US Patent 20,180,304,584.

- Effects of ERNiCr-3 butter layer on the microstructure and mechanical properties of API 5L X65/AISI304 dissimilar joint, Journal of Manufacturing Processes, Vol. 50, 2020, 305-318.

4-Explain if the stress-strain curves are engineering or true one.  

5-The results of table 3 should be with standard deviations.

6-The condition of the tensile test (environment temperature, strain rate ….) should be mentioned.

7-Ductility was reduced after the aging in the tensile test. However the authors did not discuss this phenomenon. This should be addressed.

Author Response

The paper contains some interesting results that make it publishable after the following mandatory revisions:

1-The title of the paper is incomplete. Details such as type of the interlayer should be mentioned in the title of the paper.

Answer: It was revised, new title is “Study on the Design, Preparation and Performance Evaluation of Heat-resistant Polyimide Resin-based Interlayer Neutron Shielding Material”.

2-Abstact is incomplete. Extra sentences should be deleted. The parameters investigated and the tests used and the general results should be motioned in this section.

Answer: It was revised according to the comments.

3-In the introduction of the paper, layer deposition processes and their effect on the properties of the layer should be discussed. The current introduction is incomplete. To modify this section, the following references can be used:

- Deposition of ceramic nanocomposite coatings by electroplating process: a review of layer-deposition mechanisms and effective parameters on the formation of the coating, Ceramics International, Vol. 45 (17), 2019, 21835-21842.

- CLAD ALUMINUM ALLOY PRODUCTS AND METHODS OF MAKING THE SAME, US Patent 20,180,304,584.

- Effects of ERNiCr-3 butter layer on the microstructure and mechanical properties of API 5L X65/AISI304 dissimilar joint, Journal of Manufacturing Processes, Vol. 50, 2020, 305-318.

Answer: It was revised according to the comments.

4-Explain if the stress-strain curves are engineering or true one.

Answer: The stress-strain curves are engineering

5-The results of table 3 should be with standard deviations.

Answer: It was revised according to the comments.

6-The condition of the tensile test (environment temperature, strain rate ….) should be mentioned.

Answer: It was revised according to the comments. The condition of the tensile test (environment temperature, strain rate ….) are mentioned in the corresponding position. The compression rate of the experiment is 1 mm / min. environment temperature is 10 ℃。

7-Ductility was reduced after the aging in the tensile test. However the authors did not discuss this phenomenon. This should be addressed.

Answer: It was revised according to the comments. This phenomenon was discussed in the corresponding position.

Reviewer 2 Report

It is ok. 

The font size of Figure No. 3 can be increased.

Conclusion chapter may be detailed with introduction

  • the results are interpreted appropriately and all conclusions are justified and supported by the results.
  •  The article is written in an appropriate way and the data and analyses are presented appropriately.
  • The data are robust enough to draw the conclusions.

Author Response

Reviewer’s comments

1.The font size of Figure No. 3 can be increased.

Answer: Thank you for your valuable comments on the paper, which is very helpful to improve the quality of the article. The font of the Figure in the manuscript was appropriately modified according to the opinions

2.Conclusion chapter may be detailed with introduction

Answer: It was revised according to the comments.

  • the results are interpreted appropriately and all conclusions are justified and supported by the results.
  • The article is written in an appropriate way and the data and analyses are presented appropriately.
  • The data are robust enough to draw the conclusions.

Author Response

Reviewer’s comments 

I have read the paper. It is interesting and includes a novel topic. It can be recommended for publication in Materials after a modification based on the following comments,

  1. English needs serious refinement. Just correcting the abstract as;

The polymer has an excellent effect in moderating fast neutrons with rich hydrogen and carbon, which plays an indispensable role in shielding devices. As the shielding of Neutrons is typically accompanied by the generation of γ-rays, shielding materials are developed from monomers to multi-component composites, multilayer structures, and even complex structures. In this paper, based on the typical multilayer structure, the integrated design of the shield component structure and the preparation and performance evaluation of the material is carried out based on the design sample of the heat-resistant lightweight polymer-based sandwich layer. Since the polymer-based material is the weak link of heat resistance of multilayer shield, in terms of material selection and modification, the B4C/ TiO2/polyimide molded board was prepared by the hot pressing method, and characterization analysis is conducted for its structure and

properties. The results show that the ball milling method can mix the materials well and realize the uniform dispersion of B4C/ and TiO2 in the polyimide base. The prepared

B4C/TiO2/polyimide molded plate has excellent thermal properties, and its glass transition temperature and initial thermal decomposition temperature are as high as 363.6 ℃ and 572.8℃, respectively. In addition, the molded plate has good toughness, performs well in compression resistance, shock resistance, and thermal aging resistance, and can be used for along time under 300 ℃. Finally, the prepared materials are tested experimentally on an americium beryllium neutron source. The experimental results match the simulation results well.

There are lots of errors inside the texts. Please recheck the paper deeply

Answer: It was revised according to the comments 

  1. Why the font size is different? See section 2.

Answer: The font is unified according to the comments.

  1. The quality of Figures 3 and 13 is unacceptable.

Answer: Figures 3 and 13 is revised according to the comments.

  1. Conclusion is short. There should be some explanations about the methodology and process

Answer: Conclusion is revised according to the comments.

  1. Reference list does not seem complete. The literature can expand more published references on the topic

The references appeared in different formats. See 11 compared to 12, for example

Answer: Reference is revised according to the comments.

Round 2

Reviewer 1 Report

cam be accepted

Author Response

The manuscript was revised according to the comments.
